# Lipid nanoparticles with PEG-variant surface modifications mediate genome editing in the mouse retina

Milan Gautam [1], Antony Jozic [1], Grace Li-Na Su [2], Marco Herrera-Barrera[1], Allison Curtis[2], Sebastian Arrizabalaga[2], Wayne Tschetter[2], Renee C. Ryals [2] ✉ & Gaurav Sahay [1,2,3] ✉

Ocular delivery of lipid nanoparticle (LNPs) packaged mRNA can enable efficient gene delivery and editing. We generated LNP variants through the inclusion of positively charged-amine-modified polyethylene glycol (PEG)-lipids (LNPa), negatively charged-carboxyl-(LNPz) and carboxy-ester (LNPx) modified PEG-lipids, and neutral unmodified PEG-lipids (LNP). Subretinal injections of LNPa containing Cre mRNA in the mouse show tdTomato signal in the retinal pigmented epithelium (RPE) like conventional LNPs. Unexpectedly, LNPx and LNPz show 27% and 16% photoreceptor transfection, respectively, with striking localization extending from the photoreceptor synaptic pedicle to the outer segments, displaying pan-retinal distribution in the photoreceptors and RPE. LNPx containing Cas9 mRNA and sgAi9 leads to the formation of an oval elongated structure with a neutral charge resulting in 16.4% editing restricted to RPE. Surface modifications of LNPs with PEG variants can alter cellular tropism of mRNA. LNPs enable genome editing in the retina and in the future can be used to correct genetic mutations that lead to blindness.

Inherited retinal diseases (IRDs) are a rare and heterogeneous group of neurodegenerative disorders that result in the loss or dysfunction of photoreceptor cells[1]. The progressive loss of these light sensing cells can result in night blindness, color blindness and the loss of visual acuity. IRDs affect approximately 1 in every 2000 people worldwide and are associated with mutations in nearly 300 genes[2]. Currently, the only clinical treatment is Voretigene neparvovec (Luxturna, Spark), an adeno-associated virus (AAV2) gene therapy for Leber Congenital Amaurosis associated with biallelic *RPE65* mutations[3,4]. This gene augmentation strategy is only applicable to patients harboring bilallelic mutations in genes <4.8 kb, the carrying capacity of AAVs. Genome editing, which corrects the mutation within the patient's own genome, has the ability to expand therapeutic options for IRDs by correcting mutations in larger genes, such as *ABCA4* (6.8 kb) in Stargardt disease and

*USH2A* (15.6 kb) in Usher syndrome type 2[5]. Furthermore, it would allow for the alteration of monoallelic mutations associated with autosomal dominant disease in which the mutant proteins exert toxic gain-of-function or dominant negative effects[6].

The clustered regularly interspaced short palindromic repeats (CRISPR)-associated Cas9 system has achieved substantial genome editing in IRD models of autosomal dominant Retinitis Pigmentosa[7], autosomal recessive Retinitis Pigmentosa[8], and Leber Congenital Amaurosis type 10 (LCA10)[9]. Based on the LCA10 preclinical results showing an approximate 30% editing rate in non-human primates, Editas Medicine, Inc. initiated the BRILLANCE clinical trial evaluating EDIT-101 for the treatment of LCA10 (ClinicalTrials.gov ID: NCT03872479), marking the first clinical application of CRISPR-mediated genome editing in the retina[10,11]. EDIT-101 consists of an AAV5 vector delivering the Cas9 nuclease and *CEP290*-specific gRNAs

[1]Department of Pharmaceutical Sciences, College of Pharmacy, Robertson Life Sciences Building, Oregon State University, Portland, OR 97201, USA. [2]Department of Ophthalmology, Casey Eye Institute, Oregon Health & Science University, Portland, OR 97239, USA. [3]Department of Biomedical Engineering, Robertson Life Sciences Building, Oregon Health & Science University, Portland, OR 97201, USA. ✉e-mail: ryals@ohsu.edu; sahay@ohsu.edu

to photoreceptor cells by subretinal injection. A total of 14 patients were treated in the trial, including 12 adults and two children. Unfortunately, the trial is currently paused as only three out of 14 participants met the threshold of clinically meaningful visual improvements, suggesting that there is room for improvement in the efficacy of their product either through modulating the gene editing tools, or the delivery platform[4].

There are two main safety concerns associated with AAV-mediated CRISPR/Cas9: (1) the delivered DNA forms an episome in the post-mitotic retinal cells resulting in constitutive Cas9 nuclease activity and (2) reports are showing high levels of AAV vector integration into CRISPR-induced DNA breaks[12]. Our main goal is to exploit the lipid nanoparticle (LNP)-mRNA delivery system to facilitate transient expression of Cas9 and improve the safety of gene editing therapeutics for IRDs.

LNPs are a versatile platform enabling efficient RNA delivery. LNPs are safe and effective after repeated, systemic and intramuscular delivery, which has contributed to their clinical advancement[13–15]. Onpattro, a systemically delivered RNAi drug, was the first LNP−nucleic acid therapeutic approved by the FDA for the treatment of hereditary transthyretin amyloidosis[16]. The second FDA approval came with the development of the LNP-mRNA vaccine against COVID-19, recently marked a watershed moment in the field of mRNA therapeutics[17,18]. Due to this success, the field has rapidly moved to advance LNP-mRNA genome editing platforms. Intellia Therapeutics (ClinicalTrials.gov ID: NCT04601051) have shown that NTLA-2001, comprised of an LNP encapsulating mRNA for Cas9 protein and a single guide RNA targeting transthyretin (*TTR*), has led to permanent decreases in serum *TTR* protein concentrations through targeted knockout of *TTR* after a single administration[19]. This clinical trial has now been approved within the United States[20].

LNPs have a core-shell morphology; wherein the core is rich in ionizable lipids, sterols, and a nucleotide payload and the shell consist of helper lipid (DSPC), ionizable lipid, and PEG-lipid (DMG-PEG2k). While the ionizable lipid remains the most essential constituent for endosomal escape of LNPs, the other components impart stability and cellular tropism. We have previously shown that independent of the type of ionizable lipid or mol % of PEG, conventional LNPs remain restricted to the highly phagocytic retinal pigmented epithelium[13,21]. Recently, we showed that by attaching 7 mer peptides to the surface of LNPs we were able to alter cellular tropism in the retina[22]. With an understanding that surface modifications are important for retinal transfection, we wanted to investigate how various functional groups would impact LNP physiochemical properties and alter retinal transfection efficiency.

In this work, LNPs are surface modified by inclusion of PEG-lipids with carboxy-ester (LNPx), carboxylic acid (LNPz) and amine (LNPa) functional groups. LNPx and LNPz facilitate gene expression in the photoreceptors, while LNPa and unmodified LNPs retained transfection in the RPE. LNPx was further modified to allow for co-encapsulation of Cas9 mRNA and sgRNA. This led to successful genome editing in the RPE, with some editing in the Müller glia. This is the first report, to the best of our knowledge, that shows LNP-mRNA system can be developed for genome editing in the retina.

## Results

### Design and characterization of novel LNP variants

LNPs containing a Cre recombinase mRNA were generated through microfluidic mixing of ionizable lipid, helper lipids, sterol, and PEG-lipids (DMG-PEG2k) with a precise molar ratio of 50:38.5:10:1.5, respectively (Fig. 1A, Table 1). We further generated three classes of LNP variants by reducing DMG-PEG2k to 1.2% and including 0.3% of functionalized PEG. LNPa contained DSPE-PEG2k-Amine, LNPx contained DSPE-PEG2k-Carboxy-NHS and LNPz contained a DSPE-PEG2k-Carboxylic acid. As a positive control, conventional LNP with only

DMG-PEG2k was used. To formulate the novel LNP variants the % mol of PEG-lipid was kept consistent at 1.5%. (Fig. 1B, Table 1).

All LNPs showed narrow size distribution, with a hydrodynamic diameter of <90 nm, a PDI value of <0.21 and encapsulation efficiency >94.8% (Fig. 1C, D). The zeta potential of an unmodified LNP was $-3.09 \pm 0.34$ mV. LNPa had an increased particle potential to $5.3 \pm 1.1$ mV (****$p < 0.0001$). LNPx and LNPz had a negative surface potential. When comparing the surface potential to conventional LNPs, the LNPx variant exhibited a 4.2-fold decrease in surface potential ($-12.9 \pm 0.9$ mV, ****$p < 0.0001$), while the LNPz variant exhibited a 2.3-fold decrease ($-7.2 \pm 0.4$ mV, ***$p < 0.001$) (Fig. 1E). Using cryo-transmission electron microscopy we investigated the morphological changes of LNP variants. All LNPs showed spherical nanoparticles with electron-dense core, demonstrating that replacing a portion of DMG-PEG2k with functionalized PEG-lipids has no observable effect on LNP morphology, which is consistent with previous reports (Fig. 1F−I)[23].

### Cellular uptake and endosomal escape of LNP variants

To confirm the intracellular uptake of our LNP variants, we used the mouse 661w photoreceptor-like cell line. 661w cone cells were treated with LNP-Cy-5 tagged GFP mRNA variants. Using confocal microscopy, the internalization of novel LNP variants was compared to the unmodified LNP-treated group (Figs. S1A-E). At 24 hours post-transfection, we did not observe any significant differences in uptake compared to unmodified LNPs (Fig. S1F), using a more sensitive flow cytometry-based assay LNPx showed a rather small, but significant 1.64-fold increase in uptake (**$p < 0.01$), whereas other variants still shown no difference compared to unmodified LNP (Fig. S1G, H). To determine if LNP variants were toxic to 661w cells, cell viability assays were performed at 24- and 48-h after transfection (Figs. S2A, B). At 100 and 200 ng concentrations, LNPx and LNPz did not alter cell viability (>94%), but at a 200 ng dose, LNPa did show cell toxicity which could be due to the positive surface charge[24]. At 24 h, cell viability was $85.5 \pm 5.6\%$ and at 48 h, cell viability reduced to $81.1 \pm 4.5\%$ with LNPa treatment.

Using LNP packaged Cre-mRNA we tested whether there were differences in endosomal escape. We have recently shown that a Galectin 8-GFP (Gal8-GFP) reporter system can be used to measure rare endosomal escape events[25]. In these cells, Gal8 recruitment, a hallmark of endosomal damage, is visualized by GFP puncta. GFP puncta were visualized in each LNP treated group at 24 h post-treatment (Fig. S3). Compared to the PBS-treated group, we observed that all LNP-treated cells had considerable Gal8-GFP recruitment, demonstrating that all LNP variants can facilitate endosomal escape. Consistent recruitment was observed between treated groups suggesting that varying the functionalized PEG has little impact on uptake or endosomal escape in vitro.

### LNP variants deliver Cre mRNA in Ai9 mice

We subretinally delivered Cre-recombinase mRNA encapsulated LNP variants to test their ability to transfect retina in-vivo. LNP variants were injected into Ai9 mice, which stably express a floxed-stop codon upstream of a tdTomato cassette (Fig. 2A). tdTomato is only produced after the successful removal of the stop cassette by Cre recombination[26]. Fundus and confocal imaging revealed robust and selective tdTomato fluorescence signal for all LNP variants as compared to PBS treated group (Fig. 2B−F).

When analyzing the fundus images, LNPx exhibited a significant increase in tdTomato fluorescence signal by 1.45-fold compared to conventional LNP (Fig. 2G, *$p < 0.05$). LNPz showed a 1.12-fold increase, while LNPa did not display a significant difference compared to conventional LNP. Localization of tdTomato fluorescence in retinal cross-sections was examined using confocal microscopy. Consistent with previous reports[13], unmodified LNP-mediated fluorescence signal

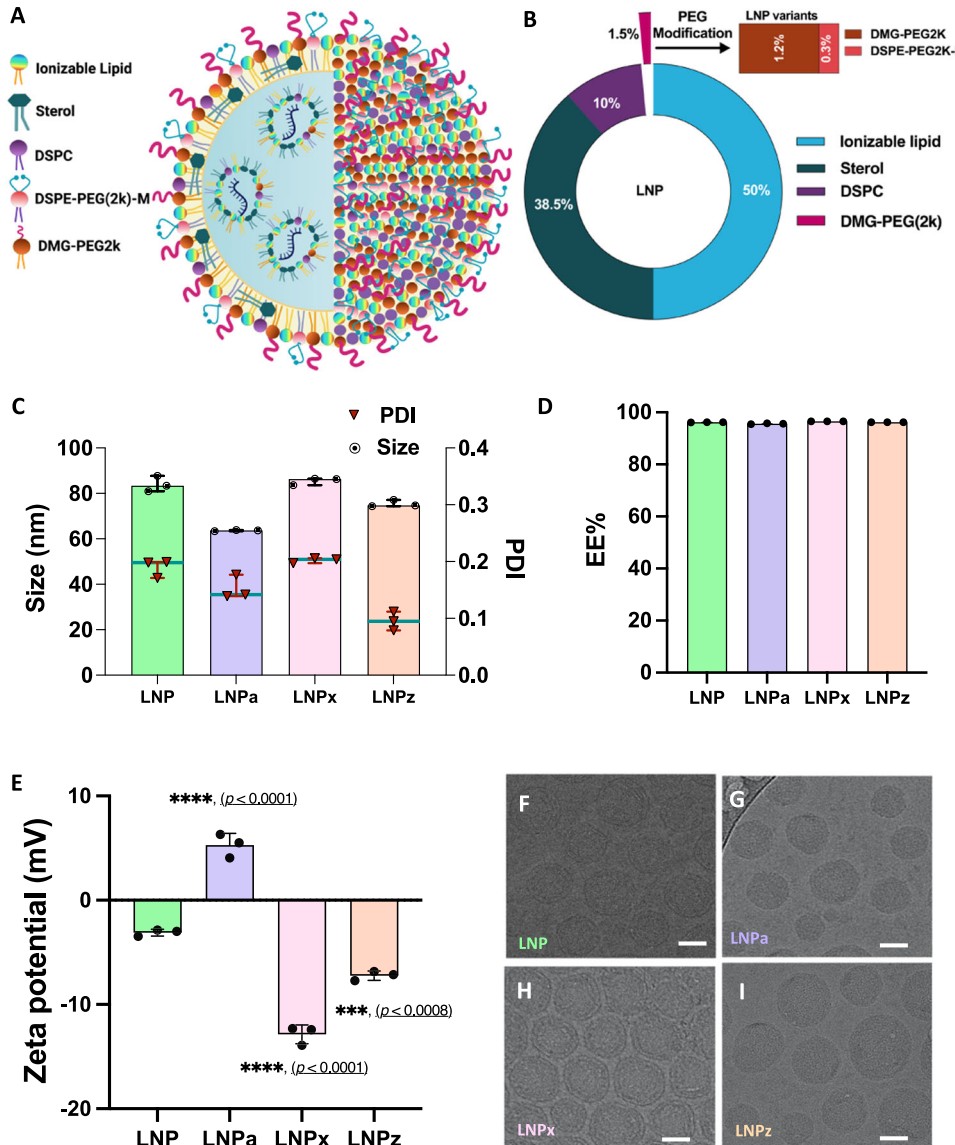

**Fig. 1 | Design and characterization of Cre mRNA encapsulated LNP variants.**
**A** Schematic of the structural organization of different LNP variants encapsulating mRNA cargo. **B** Lipid composition of LNP variants highlighting the PEG modification. DSPE-PEG2k-M stands for PEG containing different functional groups. **C** Hydrodynamic size (⊙) and polydispersity index (PDI, ▼) of Cre mRNA loaded LNP variants determined by DLS. **D** Cre mRNA encapsulation efficiency of LNP variants. **E** Surface charge of Cre mRNA loaded LNP variants determined by zeta sizer. Statistical comparison of surface charge of LNP variants were compared with unmodified LNP group. **F**–**I** Representative cryo-TEM images of LNP variants (Scale bar: 20 nm). All Data are presented as Mean ± SD. An ordinary one-way ANOVA, with Tukey's correction was used for comparison with unmodified LNP. ***$p < 0.001$, ****$p < 0.0001$ ($n = 3$ LNPs/group). Source data are provided as a Source Data file.

solely limited in the RPE (Fig. 2C). LNPa predominately mediated protein expression in the RPE, with limited localization to the photoreceptors (Fig. 2D). In contrast, LNPx and LNPz injections resulted in robust tdTomato fluorescence signal in both photoreceptors and the

RPE. With LNPx and LNPz treatment, strong tdTomato fluorescence was observed in the outer segments, inner segments, outer nuclear layer, and the synaptic region of photoreceptors (Fig. 2E, F). Quantification of tdTomato-positive photoreceptors in the ONL revealed that LNPx treatment resulted in 26.9% tdTomato-positive photoreceptors (****$p < 0.0001$), followed by LNPz with 16.5% (*$p < 0.05$), LNPa with 3.26% and unmodified LNP with 0.06% (Fig. 2H).

Hematoxylin and eosin (H&E) staining was performed to identify retinal toxicity mediated by the LNP variants (Fig. 2B–F, H & E panel). At 7 days post-injection in Ai9 mice, LNPx and LNPz variant-injected retinas showed no morphological differences to the untreated controls. Consistent with cell viability assays, retinas injected with LNPa showed disruption in the photoreceptor outer segments as well as thinning of the ONL in some treated areas, which could be due to the positive surface charge (Fig. 2D, H&E panel).

To further confirm our results of improved photoreceptor transfection, we performed immunofluorescence (IF) analysis using visual

**Table 1 | Lipid composition and surface modifications of LNP variants**

| Composition (Mol%) | LNP | LNPa | LNPx | LNPz |
|---|---|---|---|---|
| MC3 | 50 | 50 | 50 | 50 |
| DSPC | 10 | 10 | 10 | 10 |
| Cholesterol | 38.5 | 38.5 | 38.5 | 38.5 |
| DMG-PEG2k | 1.5 | 1.2 | 1.2 | 1.2 |
| DSPE-PEG-2k- Amine | 0 | 0.3 | 0 | 0 |
| DSPE-PEG-2k- Carboxy NHS | 0 | 0 | 0.3 | 0 |
| DSPE-PEG-2k- Carboxylic Acid | 0 | 0 | 0 | 0.3 |

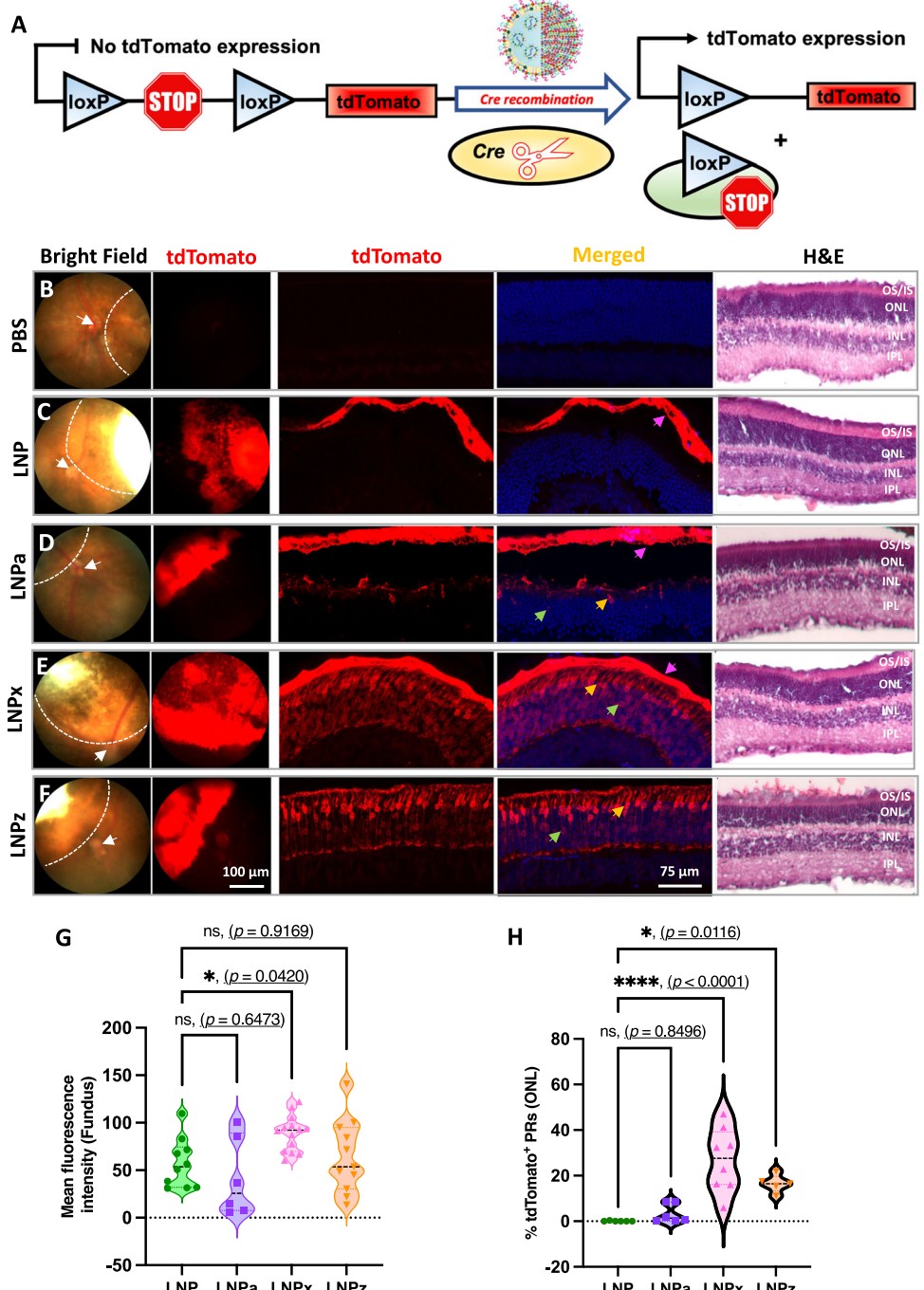

**Fig. 2 | Photoreceptor transfection efficiency of Cre mRNA encapsulated LNP variants following subretinal injection in the Ai9 mice. A** Schematic illustration of Cre dependent genetic recombination event leading to tdTomato expression. **B–F** Representative bright field and gross tdTomato expression in fundus (first two vertical panels), retinal images showing tdTomato expression in the RPE and photoreceptors (middle two vertical panels) and H&E staining (last panel) of retinal cross sections for all LNP variants. Dotted line on bright field fundus images outlines treated retina and the white arrow is pointing to the optic nerve head. Colored arrows on retinal cross section images are highlighting tdTomato expression (pink: RPE, yellow: cone photoreceptor nuclei, and green: rod photoreceptor nuclei).

**G** Quantification of tdTomato fluorescence intensity in the fundus (single eye per group, LNP = 10, LNPa = 6, LNPx = 14 and LNPz = 11). **H** Quantification of tdTomato positive photoreceptor cells in the ONL (single eye per group, LNP = 6, LNPa = 6, LNPx = 8 and LNPz = 5). 1 to 3-month-old mice were used. RPE = retinal pigment epithelium, OS/IS outer segment and inner segment, PR photoreceptor, ONL outer nuclear layer, INL inner nuclear layer, IPL inner plexiform layer. Each graph represents the Mean ± SD of each independent measurement. An ordinary one-way ANOVA, with Tukey's correction for multiple comparisons test was used for comparisons. ns-not significant, *$p < 0.05$, ****$p < 0.0001$. Source data are provided as a Source Data file.

arrestin, which labels the cytoplasm of rod and cone cell bodies as well as the entire outer segment. Staining with visual arrestin (as well as tdTomato images in Fig. 2E, F) show that LNPx and LNPz enable transfection of both rod and cones (Fig. 3A–C) and very strong localization with the outer segments. LNPx showed the highest

photoreceptor transfection efficiency as well as pan-retinal tdTomato fluorescence signal (Fig. 3D). Additional staining with recoverin, another photoreceptor marker[27], demonstrates the striking photoreceptor localization mediated by LNPx (Fig. 3E, S4). These results confirm that successful genetic recombination occurs in

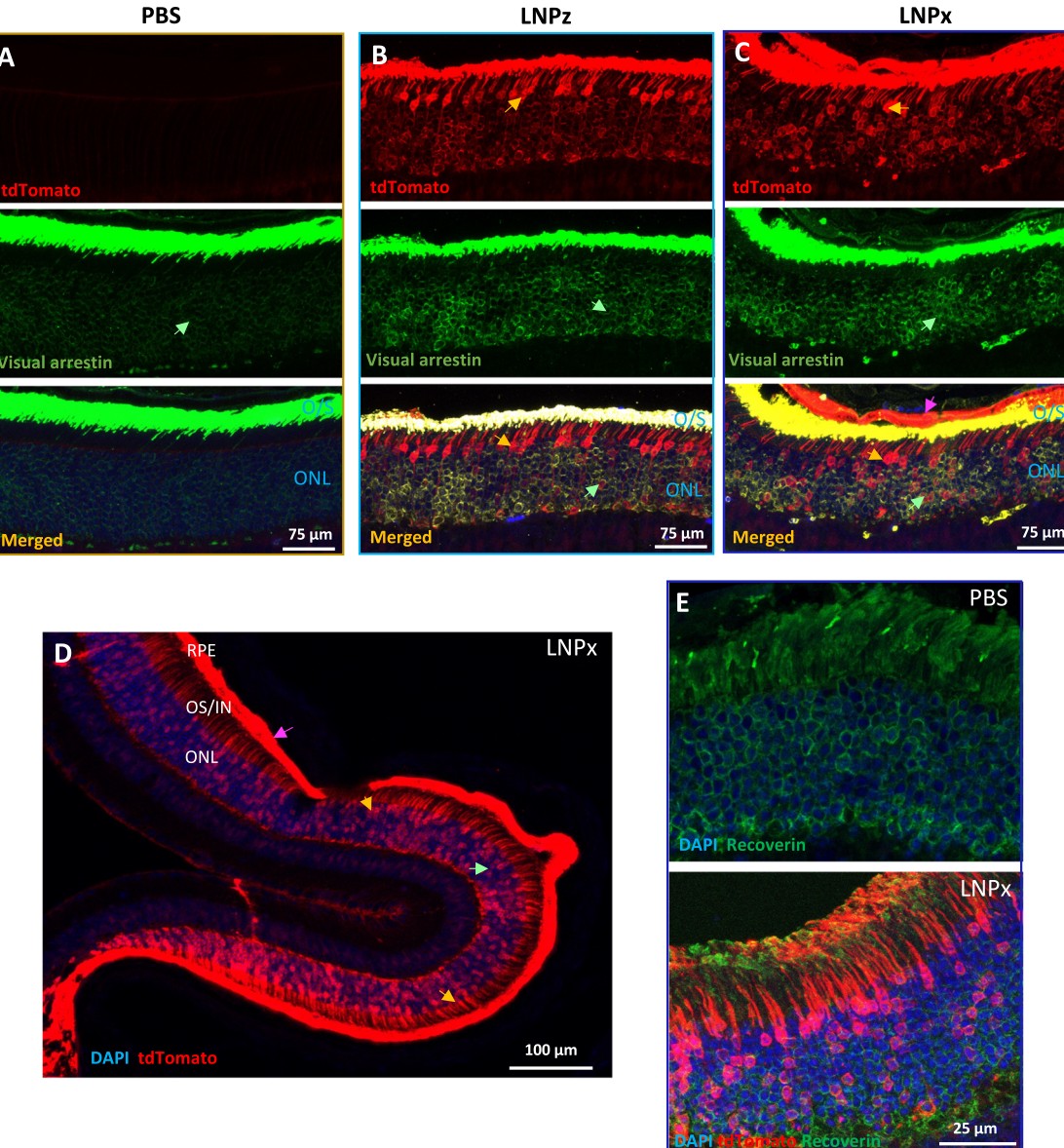

**Fig. 3 | LNP mediated photoreceptor co-localization in Ai9 retina.** Representative confocal microscopy images of the retinal sections showing tdTomato (red), stained with visual arrestin (green; rods, cones, and outer segments) and DAPI (blue) for (**A**) PBS, (**B**) LNPz, and (**C**) LNPx treatment. **D** A low magnified confocal image of an LNPx treated retina showing pan-retinal tdTomato expression from the photoreceptor synapse to the outer segment in both rod and cone photoreceptors.

**E** tdTomato fluorescence signal (red) in the photoreceptors was co-localized with recoverin (green) for PBS and LNPx treatment. 2 to 3-month-old mice were used. All representative figures from $n = 6$ single eye each group. Arrow indicates tdTomato expression (pink: RPE, yellow: cone photoreceptor nuclei, and green: rod photoreceptor nuclei). RPE retinal pigment epithelium and ONL outer nuclear layer.

photoreceptors due to carboxy-PEG modification, which has an increased anionic surface potential.

### LNP variants deliver mCherry in NRL-GFP mice model

To ensure our LNP variants could transfect photoreceptors in an additional mouse model, LNP variants encapsulating mCherry-mRNA were administered by subretinal injection in the Nrl-locus GFP-tagged mouse model (NRL-GFP), which stably expresses GFP in the photoreceptors. Exchanging the Cre mRNA for mCherry mRNA slightly altered particle size (<76 nm) and PDI value (<0.16), but there were no significant changes in mRNA encapsulation efficiency (>94.8%) (Fig. 4A, B). Interestingly, significant changes in LNP surface potential were not observed when generating mCherry encapsulated particles independent of the PEG variants, which could be due to the different cargo length (Cre−1350 nucleotides, mCherry−996 nucleotides, Fig. S5).

At 48 h post-injection, eyes were processed for immunofluorescence and confocal imaging. The PBS-injected group served as a negative control and all LNP variants were compared to the unmodified LNP (Fig. 4C–G). mCherry fluorescence signal was localized to the RPE in the LNP and LNPa groups (Fig. 4D, E), and to the photoreceptors in the LNPx and LNPz groups (Fig. 4F, G), correlating with above Ai9 findings. Quantification of mCherry fluorescence in the ONL revealed that LNPx treatment resulted in a 2.9-fold (**$p < 0.01$) increase in mCherry signal, followed by a 2.3-fold increase by LNPz (*$p < 0.05$) treatment compared to unmodified LNP (Fig. 4H). Overall, LNPx and LNPz were able to transfect photoreceptors in the NRL-GFP mice, but to a lesser extent than in the Ai9 mouse.

### In-vivo genome editing with Cas9-sgAi9 delivery

For gene editing studies, we co-encapsulated Cas9 mRNA and a loxP stop codon-targeted sgAi9 RNA in the LNPx (Cas9-sgAi9). However,

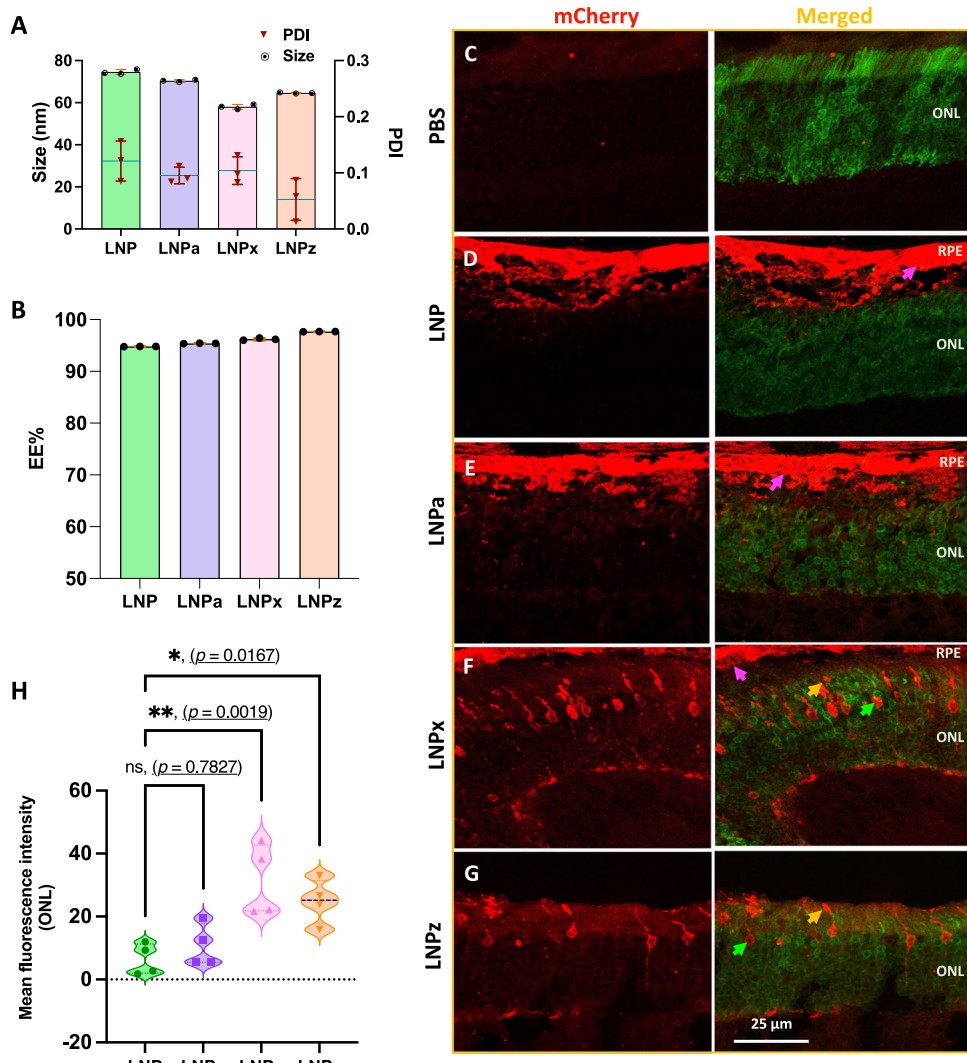

**Fig. 4 | Physicochemical characterization and in-vivo photoreceptor transfection efficiency of mCherry encapsulated LNP variants in NRL-GFP mice following subretinal injection. A** Hydrodynamic size and PDI value of LNP variants ($n = 3$). **B** mCherry mRNA encapsulation efficiency ($n = 3$). **C–G** Representative confocal images of retinal cross sections showing mCherry expression in photoreceptors and RPE (representative figures from $n = 4$ single eye per group). **H** Quantification of mCherry expression in outer nuclear layer determined by using Fiji ImageJ software ($n = 4$ single eye). Arrow highlights mCherry expression. 3- to 6-month-old mice were used. (pink: RPE, yellow: photoreceptor outer segment, and green: photoreceptor nuclei). RPE: retinal pigment epithelium, ONL: outer nuclear layer. All data are presented as Mean ± SD. An ordinary one-way ANOVA, with Tukey's correction for multiple comparisons test, was used for comparisons. ns− not significant, *$p < 0.05$, **$p < 0.01$. Source data are provided as a Source Data file.

the LNPx variant containing the DLin-MC3-DMA ionizable lipid with unmodified guides led to retinal toxicity (Fig. S6B). We chose to exchange the DLin-MC3-DMA with LP01, a novel ionizable lipid used in clinical trials for gene editing in the liver[28]. Thus, our LNPx formulation was changed to 45.5 mol% LP01, 9 mol% DSPC, 44 mol% cholesterol, 1.2 mol% DMG-PEG2k and 0.3 mol% DSPE-PEG(2k) Carboxy NHS. To increase in-vivo editing activity, stop codon-targeted sgAi9 was chemically modified based on previously published work[28] (Fig. S6A). This guide was generated to facilitate CRISPR-Cas9 mediated deletions of two repeat cassettes, which would be sufficient to activate downstream tdTomato expression[29]. The RNA cargoes (Cas9:sgAi9) were co-encapsulated at a 1:1 weight ratio. The formulation showed a high encapsulation efficiency (>97.0%) and particles were less than 68 nm in size with a 0.14 PDI value (Fig. 5A). With the ionizable lipid modification and payload changes, the resulting LNP had a nearly neutral surface potential (−0.59 ± 0.5 mV). Cas9-sgAi9 showed distinctly different morphology from previously observed LNPs, having oval or elongated protrusions extending from their outer surface (Fig. 5B). Co-

encapsulated LNPx was subretinally injected into Ai9 mice and compared with Cas9-sgGFP encapsulated LNPx (Cas9-sgGFP) as a non-targeted control group. A schematic of our in-vivo gene editing study and CRISPR-Cas9 mediated editing mechanism is presented in Fig. 5C.

At 7 days post-injection, the Cas9-sgAi9 fundus imaging revealed a significantly higher gross tdTomato fluorescence signal compared to the Cas9-sgGFP group, with a fold change of 1.97 (**$p < 0.01$) (Fig. 5D, E). Upon necropsy, RPE and neural retina flat mounts were generated. Imaging and quantification of editing events in RPE retinal flatmounts resulted 16.4% tdTomato positive RPE cells within the treated area compared to the control group (****$p < 0.0001$) (Fig. 5F, G, and S6C). The examination of neural retina flatmounts and retinal cross-sections confirmed the editing of Müller glia cells (Fig. S6D–E). A low level of editing was observed in photoreceptors, typically ranging from 2 to 3 cells per eye (Fig. S6F, G).

To strengthen the findings, next-generation sequencing (NGS) techniques were employed by extracting genomic DNA from RPE cells.

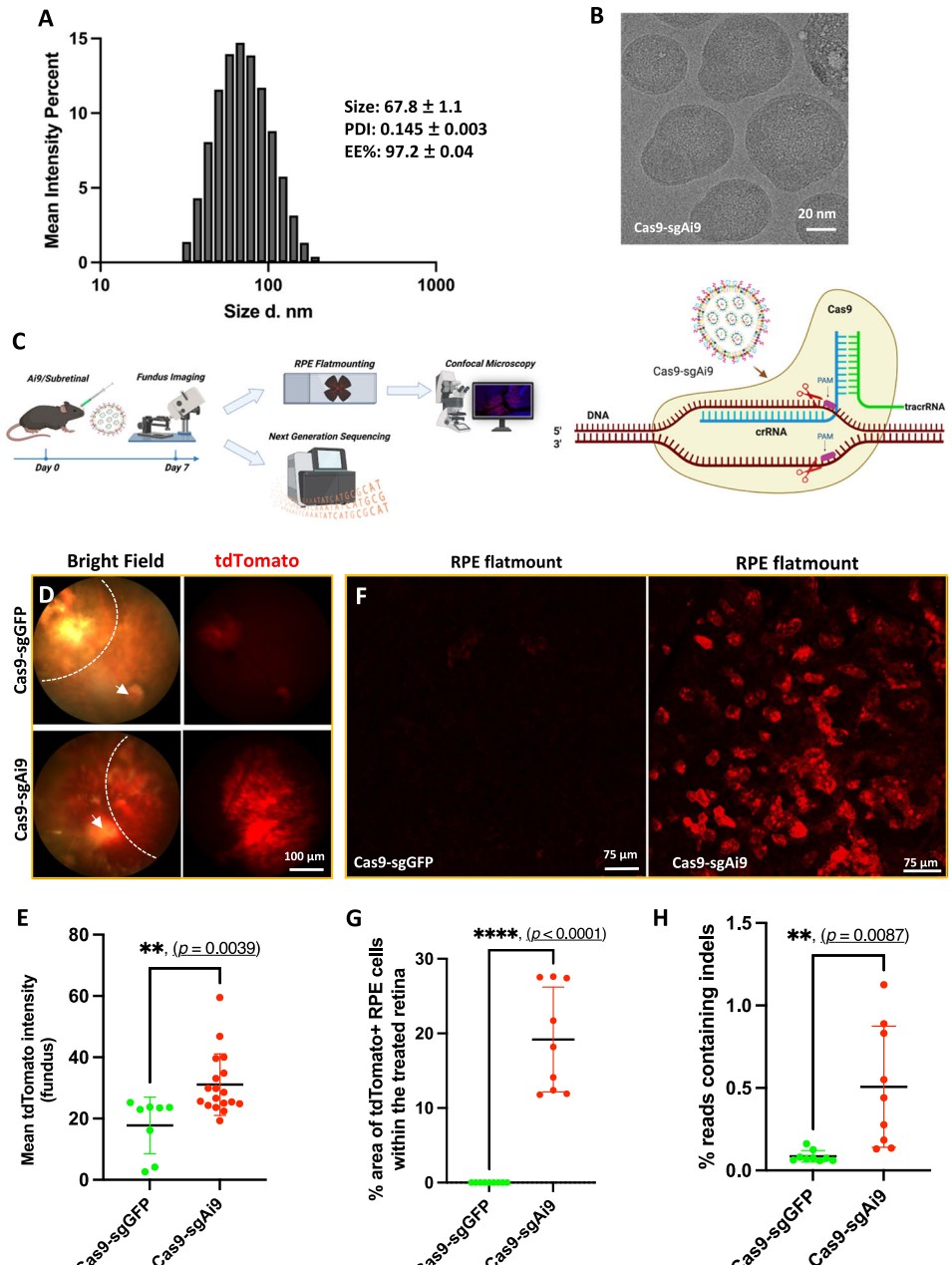

**Fig. 5 | LNPx mediated gene editing in the Ai9 retina. A** Characterization of Cas9-sgAi9 encapsulated LNPx using DLS (inset: Size, PDI and EE%) and (**B**) cryo-TEM showing oval shape structure (representative figure from *n* = 3 replicates). **C** Schematic illustration exhibiting delivery of LNPx via subretinal route, retinal imaging and processing as well as CRISPR-Cas9 editing mechanism. The figures were created with BioRender.com. **D** Representative fundus imaging showing editing events mediated by Cas9-sgAi9-LNPx. Dotted line outlines treated retina and white arrow is pointing to the optic nerve head. **E** Quantification of tdTomato signal in fundus images. Cas9-sgAi9-LNPx (*n* = 18 single eye) mediated tdTomato expression at significantly increased levels compared to the control non-targeted Cas9-sgGFP-LNPx (*n* = 8 single eye). **F** Representative flatmount images showing gene editing events in the RPE. **G** Quantification of gene editing events in the RPE. The area of tdTomato expression within the treated retina was quantified using ImageJ (*n* = 9 single eye per group). **H** Quantification of editing events by next-generation sequencing showed increased edits in the Cas9-sgAi9. RPE tissue was harvested for NGS. Cas9-sgGFP-LNPx was used as a non-targeted control group. All presented data were obtained from particles containing LP01 and LNPx (*n* = 9 single eye per group). All samples were injected subretinally. 2 to 4-month-old mice were used in the above experiments. All data are presented as Mean ± SD and two-tailed Welch's *t* test was used for comparisons. **p < 0.01, ****p < 0.0001. Source data are provided as a Source Data file.

This allowed for the quantitative determination of the total editing percentage in the RPE from high-quality NGS data (Figure S7). The Cas9-sgAi9 group exhibited 0.51% of reads containing indels, confirming a 5.94-fold higher RPE editing compared to the control group (Fig. 5H). Overall, the results indicated that Cas9-sgAi9 delivered by LNPx mediated gene editing in the retina, with the most robust editing observed in the RPE.

## Discussion

Although CRISPR-Cas9 editing technologies continue to show great promise for the correction of genetic diseases, optimization of different delivery systems remains to be thoroughly explored. For instance, in the case of EDIT-101, although it represented a first-in-human CRISPR-Cas9 application, the clinical trial is currently paused[10]. Safety was shown at all dose levels, but efficacy was limited as only 3

out of 14 participants met the threshold of clinically meaningful visual improvements[4]. For this phase 1/2 clinical trial, an AAV5 vector delivers the Cas9 nuclease and guide RNAs as single-stranded DNA to the post-mitotic photoreceptor cells. Within the nucleus, the DNA becomes double-stranded and remains as an episome. With available promoter activity, the gene editing machinery can be continually transcribed for the life of the cell. Two major safety concerns, which have not been fully addressed, are (1) off-target editing/unwanted indels and (2) AAV integration into Cas9 induced double-strand breaks. CRISPR-Cas9 systems are known to have a low percentage of off-target editing regardless of continued improvements of the system. It remains unclear how these constitutive off-target cuts will impact the genome over the lifetime of the cell. Furthermore, it remains unclear if non-homologous end-joining (NHEJ) and microhomology-mediated end joining (MMEJ) repair mechanisms which usually have high efficiency in-vivo, remain intact in degenerating retinal cells. Of greater concern, multiple studies have demonstrated high levels of AAV integration (up to 47%) into the targeted CRISPR-Cas9 cut sites both in vitro and in vivo.

With an LNP-mRNA system, both the Cas9 nuclease, encoded as mRNA, and the guide RNA are encapsulated through microfluidic mixing. These components are delivered to the cytoplasm of the cell, where the Cas9 mRNA is translated into protein that the guide RNA binds to. The guide RNA-Cas9 ribonucleoprotein complex then shuttles into the nucleus to find the target site in the genome. LNP-mRNA systems allow for transient CRISPR-Cas9 enzyme activity, which decreases the concern of constitutive off-target cutting[30]. Additionally, there is no chance of gene editing machinery integration in the genome. Another benefit consists of the ability to perform repeat dosing of LNPs, as seen with ONPATTRO[31] and the COVID-19 vaccines[32], which will no doubt improve editing efficiency. Based on these improvements in safety and the potential for multiple dosing to increase gene editing efficiency, we will continue to engineer LNP-mRNA systems that enable gene editing in the retina.

Initially, anionic PEG-lipid (LNPx-DSPE-PEG2k-Carboxy-NHS and LNPz-DSPE-PEG2k-Carboxylic acid) surface modifications generated a significant change in the LNP surface potential. It is worth noting that in LNPx formulation, NHS esters are pH-sensitive and can hydrolyze to form carboxylic acids. At a stable reaction condition, these esters can also be used to conjugate proteins, antibodies, peptides, and other substances that can efficiently increase gene editing efficiency[33]. It is therefore possible that LNPx and LNPz formulations are similar and display carboxylic acids on their surface in vivo resulting in robust photoreceptor transfection. In the subsequent experiment, when the LNP cargo was exchanged from Cre mRNA to mCherry mRNA, the magnitude of the negative surface potential decreased, and photoreceptor transfection decreased as well. In the final experiment, both the cargo (Cas9 + sgAi9) and the ionizable lipid (LP01) were changed, generating a neutral particle and almost a complete loss of photoreceptor transfection was observed. Our data provides evidence that an anionic charge leads to photoreceptor transfection, however, we realize there are several caveats. Along with the different cargo, two different mouse models were used, the Nrl-GFP being a less sensitive model than the Ai9 reporter model. Regarding, LNPx containing the Cas9 mRNA-sgAi9, these formulations had an abnormal structure as compared to the spherical core-shell morphology of the regular LNPs, which may also have led to changes in cellular tropism and decreased gene editing of photoreceptors. We speculate that LNPs with the carboxylated surface modification have a modified protein corona that enable entry in the photoreceptors.

One main limitation of our carboxylated LNPs was that they were not able transfect the photoreceptors with gene-editing cargo. However, we show two separate studies suggesting that surface modifications are key to altering cellular tropism in the retina[21,22]. Thus, we speculate that with continued optimization of surface modifications,

potentially with the use of alternative PEG variants (polysarcosine or 2-phenyl oxazoline) and additional 7-mer peptides, we can achieve delivery of gene editors to the photoreceptors. Future studies evaluating large libraries of LNPs encapsulating Cas9 mRNA via DNA- or mRNA-barcoding techniques will elucidate LNP physiochemical properties that allow for photoreceptor delivery of gene editing cargo[34,35].

While superior LNPs can lead to improved gene transfection, it is of utmost importance to optimize gene editing guides and delivery techniques for enhanced gene editing. There have been numerous studies that have chemically modified the sgRNA to enhance its stability without compromising its efficiency[36]. Partial modification of folding motifs in the sgRNA with 2′-O-methylation coupled with phosphorothioate bonds at the terminal ends of sgRNA was found to significantly increase in-vivo editing efficiency[28]. We also observed increased gene editing events and decreased retinal toxicity following sgAi9 modification. Additionally, increasing the number gene editing treatments with repeat injections will allow for increased editing events[37]. While repeat subretinal delivery may not be feasible in humans, the suprachoroidal injection, which is an outpatient procedure, has become an attractive injection method for the back of the eye[38]. Future studies assessing the immune response and transfection efficiency of LNPs after suprachoroidal delivery is still needed. Intravitreal delivery, which would enable multiple dosing and easy administration of these new genetic medicines, also remains highly desirable.

Our goal is to develop translatable LNP-mRNA gene editing systems for IRDs, such as LCA10, autosomal dominant RP, RPE specific mutations and Ushers Syndrome. To the best of our knowledge, this is the first report demonstrating LNP-based gene editing within the retina. Thus, laying the foundation for developing safer gene editing platforms for IRDs with the use of LNPs. With continued optimization of LNP surface modifications and gene editing cargo, the development of therapeutic gene editing systems will expand.

## Methods

### Materials

Dlin-MC3-DMA was purchased from BioFine International Inc. (BC, Canada). LP01 (BP-Lipid 215) was purchased from BroadPharm. 1,2-distearoyl-sn-glycero-3-phosphocholine (18:0 PC, DSPC), 1,2-distearoyl-sn-glycero-3-phosphoethanolamine-N-[carboxy(polyethylene glycol)−2000, NHS ester] (DSPE-PEG2k-Carboxy NHS), 1,2-distearoyl-sn-glycero-3-phosphoethanolamine-N-[carboxy(polyethylene glycol)−2000] (DSPE-PEG2k-Carboxylic acid), 1,2-distearoyl-sn-glycero-3-phosphoethanolamine-N-[amino(polyethylene glycol)−2000] (DSPE-PEG2k-Amine) was purchased from Avanti Polar Lipids. Cholesterol and 1,2-dimyristoyl-rac-glycero-3-methoxypolyethylene glycol-2000 (DMG-PEG2k) were purchased from Sigma Aldrich. To encapsulate cargo into LNPs, Cre mRNA (5moU) - (L-7211), mCherry mRNA (5moU) - (L-7203), Fluc mRNA (5 moU) - (L-7602), Cyanine 5 EGFP mRNA (5moU)-L-7701 and Cas9 mRNA (5moU)- (L-7206) were purchased from Trilink Biotechnologies. sgAi9 was custom synthesized from Integrated DNA Technologies. Anti-mCherry pAb (Cat. # NBP2-25157) was purchased from NOVUSBIO and Anti-Recoverin (Cat. # AB5585) was purchased from Millipore Corp. USA.

### Cell Culture

All cell culture media and supplies were obtained from Thermo Fisher Scientific (Waltham, MA). 661 W cone cells were generously provided by Prof. Muayyad Al-Ubaidi, University of Houston, Houston, TX. 661w cells were cultured in Dulbecco's modified Eagle's medium (DMEM, Cat. #11965175) supplemented with 10% heat-inactivated fetal bovine serum (FBS) (Hyclone Laboratories Inc., Logan, UT), 1× penicillin/streptomycin (Thermo Fisher, Federal Way, WA), 23 mg/l Putrescine, 40 µl of β-Mercaptoethanol, 300 mg/l glutamine, and 40 µg of hydrocortisone 21-hemisuccinate and progesterone. Previously developed[25]

Gal8-GFP reporter cells using Human Embryonic Kidney 293T/17 cells (CRL-11268; ATCC, Manassas, VA) were also cultured in DMEM supplemented with 10% heat-inactivated FBS and 1% penicillin/streptomycin. All the cells were maintained in an atmosphere of 5% $CO_2$ at 37 °C.

## Animals

Breeder Ai9 (Strain # 007909) mice were purchased from The Jackson Laboratory (Bar Harbor, ME, USA). Ai9 is a Cre reporter tool designed to have a loxP- flanked STOP cassette preventing transcription of tdTomato under the control of a ubiquitous promoter. Following Cre-mediated recombination, Ai9 mice express robust tdTomato[39]. Breeder NRL-GFP mice were generously endued by Dr. Anand Swaroop. These transgenic reporter's strain of origin is (C57BL/6 x SJL)F2. Both male and female mice were used in the study. Male and female mice were pooled and tested collectively throughout the study. The study did not involve separate analyses for each gender, and there was an unequal distribution of male and female mice. No animals or data points were excluded during the experiment or data analysis. All mice were housed in a specific-pathogen-free animal facility at ambient temperature (22 ± 2 °C), air humidity 40–70% and 12-h dark/12-h light cycle. All mice were maintained on a free access to standard rodent chow diet (5L0D - PicoLab) and water. All the mice used in the experiments were 1 to 6 months old and were bred in-house for the experiments. Mice were euthanized by carbon dioxide inhalation at a flow rate of 2 to 5 liters per minute. No masking or randomization was used in the study.

## Ethics statement

All experimental procedures were carried out in accordance with the protocols approved by Oregon Health & Science University's Institutional Animal Care and Use Committee (IP00001707 and IP00000610) and in accordance with the Association for Research in Vision and Ophthalmology's (ARVO) Statement for the Use of Animals in Ophthalmic and Vision Research[13].

## Formulation of mRNA LNPs and characterization

To obtain the desired molar ratio in the organic phase, Cre or mCherry mRNA encapsulated LNPs were prepared using DLin-MC3-DMA/DSPC/ Cholesterol/DMG-PEG2k and/or functional PEG lipids in the mol percent ratio of 50/10/38.5/1.2/0.3 in the ethanol solution. Before microfluidic mixing in the NanoAssmblrTM (Precision NanoSystem) at a 1:3 flow ratio, mRNA was dissolved in 50 mM citrate buffer pH 4.0. Following formulation, LNPs were diluted in sterile PBS (pH 7.4) and dialyzed against 3 liters of PBS for 4 hours at room temperature using a 10 kDa Slide-A-Lyzer dialysis bag, before being transferred to fresh PBS solution overnight. To prepare the Cas9 and sgRNA encapsulated LNPx, lipid composition was changed to 45.5/9/44/1.2/0.3 using biodegradable LP01 ionizable lipid/DSPC/Cholesterol/DMG-PEG2k/DSPE-PEG-2k-Carboxy NHS, respectively. The RNA cargoes were dissolved in 50 mM acetate buffer (pH 4.5), formulated using NanoAssmblrTM, and dialyzed in pre-cooled PBS buffer (pH 7.4). The remaining steps are the same as those in the above-described procedure. LNPs were concentrated after dialysis using a MWCO, 10 kDa Amicon Ultra-4 mL centrifugal filter unit for in-vitro and in-vivo studies (Millipore, Burlington, MA). LNP variants' size distribution, PDI and surface potential were determined using Dynamic Light Scattering (DLS) in the Zetasizer Nano ZS (Malvern Panalytical Inc., Westborough MA) at 25 °C. Quant-iT RiboGreen RNA reagent was used to estimate mRNA encapsulation efficiency and concentration.

## Cryo-TEM image acquisition and processing

Falcon III and K3 Summit cameras with DED at 300 kV were used to capture cryo-TEM images. The Vitrobot Mark IV system (FEI) was used to plunge-freeze a copper lacey carbon film-coated Cryo-EM grid

(Quantifoil, R1.2/1.3 300 Cu mesh). 2 µL of LNP was dispensed onto the glow discharged grids in the Vitrobot chamber maintained at a temperature of 23 °C and a relative humidity of 100% to freeze the samples. The sample was incubated for 30 seconds before being blotted with filter paper for 3 seconds before being submerged in liquid ethane cooled by liquid nitrogen. The frozen grids were meticulously examined for any defects, clipped, and assembled into cassettes[23,40]. The images were taken at an electron dose of 15-20 e − /Å2 using 45,000 nominal magnifications with 1.5 binning then processed and analyzed using Fiji (Fiji ImageJ2 version: 2.9.0/1.53t).

## Cellular uptake of Cy-5 tagged LNP variants

Cy5- tagged LNP variants were used to study the cellular uptake in 661w cone cells. 661w cells were seeded at a density of 60,000 per well in an 8-well µ-Slide (Ibidi, Fitchburg, WI) for confocal microscopy. All cells were treated with Cy5-tagged LNP variants in a dose equivalent to 100 ng/well of mRNA cargo, incubated for 24 hours at 37 °C, washed with 1x DPBS, and fixed for 7 minutes at room temperature with 4% paraformaldehyde. Cells were then washed twice with 1x DPBS, stained with DAPI (0.75x), washed again, and mounted with Fluoromount-G Mounting Medium (Thermo Fisher cat. # 00-4958-02) and imaged using confocal microscope (Leica DMi8, Leica Microsystems). Same exposure parameters were set for the PBS and LNP variants treated groups. The fluorescence intensity of images was analyzed using Fiji ImageJ2 (version 2.9.0/1.53t; National Institutes of Health, Bethesda, MD).

Flow cytometry was used to confirm cellular uptake of the particles. Approximately 160,000 661w cells were seeded per well in 6 well plates and incubated at 37 °C for 24 h in a 5% $CO_2$ atmosphere. Cells were treated with 100 ng/well equivalents of mRNA from Cy5-tagged LNP variants and incubated for 24 hours. Following incubation, cells were washed twice with flow cytometry staining buffer (eBioscience, cat. # 00-4222-26), tyrpsinized using Trypsin (0.25%)-phenol red (Gibco, cat. # 15050065), harvested and re-suspended in the FACS buffer then flow cytometry analyzed (10000 cells count, BD Biosciences). A 670/30 bandpass filter was used to detect the Cy-5 signal. Data were processed using FlowJo 10.8.1.

## In-vitro cell viability

A cell density of 4,000 per well was plated in white, clear-bottom 96-well plates and allowed to grow to 60-70% confluency for 24 hours at 37 °C. 661w cells were treated with LNP variants encapsulating Fluc mRNA at concentrations of 100 and 200 ng in the medium. All the cells were incubated for an additional 24 and 48 hours and cell viability (Promega CellTiter FluorTM Cell Viability Assay) was measured.

## Endosomal escape study

The HEK 293 T/17 Gal8-GFP reporter cells at a density of 60,000 per 8-well coated with poly-D-lysine (Thermo Fisher, Waltham, MA) were incubated overnight to attach the cells. Next day, LNP variants were added to the cells and incubated for 24 hours. After that media was aspirated, cells were washed with 1x DPBS and fixed with 4% paraformaldehyde for 7 minutes at room temperature. After fixing, cells were washed (3 times) with 1x DPBS, DAPI stained (1x), washed (3 times) and mounted using Fluoromount-G and cover slipped. Images were taken by Leica DMi8 confocal microscope.

## Subretinal injection

Before the subretinal injection, mouse eyes were dilated by topical administration of 0.5% proparacaine, 1% tropicamide, and 2.5% phenylephrine and underwent general anesthesia with a ketamine (100 mg/kg)/xylazine (10 mg/kg) cocktail administered intraperitoneally. To initiate the injection, 2.5% hypromellose was placed over the eye and a 30-gauge needle was used to make an incision in the limbus. After that, a glass coverslip was placed over the eye to allow

visualization of the retina. Using a Hamilton syringe with a 33-gauge blunt needle, 1 µL of PBS or Cre-LNP variants (300 ng mRNA per eye) or mCherry-LNP variants (500 ng mRNA per eye) or Cas9-sgTdTOM LNPx (200 ng total mRNA per eye) were delivered to the subretinal space. To visualize the temporal retinal detachment, a 2% fluorescein solution was added to the PBS and LNP variants.

## Fundoscopy

Live, in-vivo retinal imaging was performed with the Micron IV (Phoenix Research Laboratories, Pleasanton, CA). To observe general retinal health, bright field images were acquired. All fundus imaging were conducted with the same program settings. To capture tdTomato, a 534/42 nm BrightLine® single-band bandpass filter (Semrock, Rochester, NY) was used. Light intensity, exposure and gain were kept consistent across all RFP images. Mean tdTomato fluorescence intensity was measured using the entire retinal image of the fundus. Mean fluorescence obtained from all the LNP-treated groups were subtracted from PBS treatment group to get net fluorescence intensity.

## Tissue preparation

LNP-Cre-injected Ai9 mice were harvested 1-week post-injection, whereas LNP-mCherry-injected NRL-GFP mice were harvested 48 hours post-injection. The limbus of all injected eyes was marked at the 12 o'clock position with a hot needle to aid orientation. Under dim light, eyes were enucleated and immediately placed in 4% paraformaldehyde overnight. After fixation, whole globes were rinsed in PBS and then incubated in 30% sucrose for 2 hours at 4 °C before embedding in cryostat compound (Tissue-Tek O.C.T. Compound, code # 4583). The embedded eyes were snap-frozen in a dry ice bath. Eyes were sectioned at 12 microns with a cryostat (Microtome HM550; Walldorf, Germany) and stored at −20 °C. The eyes designated for flatmount analysis were prepared following previous protocol[41]. Eyes were burn-marked on the superior cornea, enucleated, cleaned, and cornea punctured. Eyes were immersion fixed in 4% PFA at 4 °C overnight, followed by an immersion 30% sucrose incubation at 4 °C for 1 week. Two microscope systems, the Leica M60 and Leica Stereo S6 were employed during flatmount dissections. During dissection, the eyes were kept moist with approximately 30 µL 1x PBS and forceps were used to manipulate the globe. Scissors were inserted into the corneal puncture and cut around the cornea and limbus, removing the cornea, limbus, lens and vitreous, without disturbing the retina or RPE. To ensure proper orientation, a triangular notch was cut into the eye cup at the superior corneal burn mark. Using forceps, the intact whole neural retina was grasped and peeled away from the ciliary body, then lifted and removed by going under the retina to grasp the optic nerve head and lifting the retina off and placing in PBS. Four symmetric radial cuts to the optic nerve head, manifesting as petals, were made on the remaining eye cup and the petals were carefully spread-out using forceps.

## Immunofluorescence (IF) and Confocal microscopy

Retinal cryosections were washed three times in 1x PBS. After incubating samples in 0.3% Triton X-100 for 1 h at room temperature, they were blocked in a Super blocking buffer (Thermo Scientific, Cat. # 37515) for 1 hour. Slides were incubated overnight at 4 °C in primary antibodies, which were diluted in 0.1% Triton X-100 Super blocking buffer. Following primary antibody incubation, retinal sections were washed three times with 1x PBS before being incubated with secondary antibody for 1 hour at room temperature. All retinal sections were counter-stained with DAPI for 5 minutes at room temperature. After final rinsing, retinal sections were cover-slipped with Fluoromount-G. For Ai9 cryosections, anti-recoverin rabbit antibody (1:500, Sigma-Aldrich, Cat. # AB5585) and corresponding goat anti-rabbit Alexa Fluor 700-IgG (H + L) (1:1000, Invitrogen Cat. # A21038) were used for

detection. Visual arrestin antibody (E-3, 1:100, Cat. # SC-166383) and corresponding Alexa Fluor Plus 488-IgG (H + L) (1:200, Cat. # A21038) was also used to confirm photoreceptors co-localization. For NRL-GFP mice cryosection, anti-mCherry rabbit antibody (1:250, Novus Biologicals, Cat. # NBP2-25157) and corresponding donkey anti-rabbit Alexa Fluor 594-IgG (H + L) (1:500, Invitrogen, Cat. # A-21207) were used for detection.

Retinal sections were analyzed by confocal microscope equipped with confocal microscope (Leica DMi8, Leica Microsystems). All the images were captured with identical exposure setting at either 10x or 40x or 60x magnification using Z-stacks (spanned 10 µm with 1 µm interval). Maximum fluorescence projections were used for further analysis. For Cre mRNA studies, the number of tdTomato-positive photoreceptor nuclei in the ONL were counted and divided by the total number of DAPI-stained photoreceptor nuclei in 40x images of retinal sections, providing a percentage of tdTomato-positive photoreceptors for each group. For the NRL-GFP experiment, using ImageJ software, the mean fluorescence intensity of mCherry was measured in a region of interest (ROI), the ONL layer of the retinal sections using 40x images. To calculate the editing events in the RPE flatmounts, several areas within the treated RPE of each eye were imaged using the Leica confocal microscope. The regions of tdTomato expression within the treated RPE were traced using the polygon selections tool and the ROI manager tool in ImageJ. ImageJ was then used to quantify the area of tdTomato expression (converted from pixel measurements). Then the total tissue area was measured using the same tool. Pixel area measurements were converted to µm units using the scale bar provided in each image. The percentage of tissue area expressing tdTomato within the treated retina was calculated for each image and then averaged for each treatment.

## Hematoxylin and eosin staining of retinal cross-sections

Retinal tissues were first stained with hematoxylin for 30 seconds at room temperature and then rinsed in tap water. Next, retinal sections were decolorized in acid alcohol (10% acetic acid and 85% ethanol in water), followed by rinsing in tap water. In the blueing step, retinal sections were immersed in a saturated sodium bicarbonate solution, rinsed with tap water, and then immersed in 95% alcohol. Finally, staining was performed with Eosin-Phloxine for 10 seconds, followed by a dehydration step and cover slipping. All the imaging was performed using Leica optical imaging microscope using Zen 3.5 (blue edition). Images were collected using 4x objectives.

## Next generation sequencing (NGS)

NGS was performed via a two-step PCR. Two primer sets were designed to amplify the 3 stop codons in the Ai9 loxP-stop-loxP cassette (Table S2). The first primer set generates an amplicon containing the first stop codon. The second primer set amplifies both the second and third stop codons. RPE tissues were harvested 7-day post subretinal injection of either Cas9-sgGFP or Cas9-sgAi9 encapsulated LNPx. Genomic DNA (gDNA) was harvested from RPE tissues using the Qiagen DNeasy Blood & Tissue kit (Qiagen, #69506) according to the manufacturer's instructions. For the first PCR reaction: 50 ng of gDNA, 1 µL of 5 µM Ai9_NGS_F1_F or Ai9_NGS_F2_F, 1 µL of 5 µM Ai9_NGS_R, nuclease-free $H_2O$, and KAPA HiFi HotStart ReadyMix were combined in a 25 µL reaction volume and thermocycled according to the tdTom_Ai9_NGS method (Table S3). For the second PCR: 2 µL of 1st PCR reaction product, 1 µL xGen-UDI primer pair (Integrated DNA Technologies, #10005922), nuclease-free $H_2O$, and Phusion High Fidelity Master Mix (New England Biolabs, #M0531S) were combined in a 25 µL reaction volume and thermocycled according to the Ai9_2nd_PCR_NGS method (Table S4). The samples were then run on a gel, excised, pooled together, and harvested. The pooled library DNA concentration was quantified via qPCR using the KAPA Quantification Kit (Roche, #07960140001). After quantification the library was pooled and

diluted to 4 nM before being prepared for NGS using Illumina's standard denature and dilute protocol. The NGS library was run on an Illumina MiniSeq using a 300-cycle mid output kit (Illumina, FC-420-1004). Editing was quantified using CRISPResso2 (v2.2.14) in standard mode, using the indel count in the "CRISPResso_quantification_of_editing_frequency.txt" file. The insertions and deletions were added together before being normalized to the number of reads, resulting in a % of reads containing indels.

## Statistical analysis

For fundus quantification, since all images were taken at the same exposure settings, the ROI was generated around the entire retinal image to calculate the mean fluorescence intensity using ImageJ (Fiji ImageJ2 version: 2.9.0/1.53t). The number of tdTomato-positive photoreceptor cells or the intensity of mCherry fluorescence in the ONL was measured using the same tool. In each layer, a distinct ROI was outlined in the Ai9 and NRL-GFP retinal sections. To determine the relative changes, the data were normalized to the unmodified LNP group, and fold change values were calculated. Statistical analysis was conducted using an ordinary one-way ANOVA, followed by Tukey's multiple comparison test. In the gene editing section of the study, either the quantification of tdTomato in the fundus or the count of tdTomato-positive RPE cells was performed as described in the methods. Welch's $t$ test was used for statistical analysis. The statistical analysis was performed using GraphPad Prism software (version: 9.5.0). All data were presented as Mean ± SEM, and a $p$-value below 0.05 was considered statistically significant.

## Reporting summary

Further information on research design is available in the Nature Portfolio Reporting Summary linked to this article.

## Data availability

The authors declare that all other data supporting the findings of this study are available within the paper and its Supplementary Information files. Sequencing data is available from the Sequence Read Archive under accession code PRJNA991562. Source data are provided with this paper.

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

## Acknowledgements

The authors would like to thank Dr. Jeonghwan Kim, Dr. Yulia Eygeris, and Dr. Mohit Gupta for their invaluable help and support throughout the course of our experiments. We would like to express our appreciation to Dr. Anindit Mukherjee for providing valuable suggestions regarding the figures included in our study. This project was supported through funding from the National Eye Institute 1R01EY033423-01A1 (G.S.), Casey Eye Institute Core Grant P30 EY010572 from the NIH (R.C.R.), ARCS Foundation Scholar 1T32GM142619-01 (G.L.S.) and unrestricted departmental funding from Research to Prevent Blindness. Utilizing BioRender.com, certain figures were created.

## Author contributions

G.S. conceived of the idea and directed the entire research, Data curation was done by G.S. and M.G., and Research was performed by M.G., R.C.R., A.J., G.L.S., M.H.B., A.C., W.T., and S.A., Methodology was performed and optimized by M.G., R.C.R., and A.J., Data analysis was carried out by M.G., all resource and supervision was done by G.S., R.C.R., Original draft was written by M.G. and G.S. with feedback from all the authors. Final review and editing were done by M.G., R.C.R. and G.S.

## Competing interests

M.G. and G.S. are listed as co-inventors in U.S. Patent application WO2023049493A1. R.R., M.H.B., and G.S. are listed as co-inventors in U.S. Patent Application WO2023122721A2. G.S. is a cofounder of EnterX Bio. EnterX Bio has a scientific research agreement with OSU. G.S. has a conflict management plan at OSU. All other authors declare they have no competing interests.
