## [Peer Review File · Nature Communications]

REVIEWER COMMENTS

Reviewer #1 (Remarks to the Author):

In this manuscript, Sahay and co-authors conducted an exciting and comprehensive study using lipid nanoparticles (LNPs) for mRNA delivery and gene editing in the eye. The results from two different animal models showed that surface-modified LNPs, specifically those with carboxylation LNP-mRNA formulations exhibited significantly higher expression in the retina and can be applied to target multiple types of retinal cells. Additionally, in vivo results displayed that the modified LNPs encapsulating Cas9 mRNA and sgRNA can effectively induce retinal gene editing. Overall, this is a very meaningful work and an important report on modified LNPs for retinal gene editing. Below are minor comments.

1. What is the rationale to choose 0.3% of functionalized PEG as the molar ratio?
2. What is the location of the bright spot in the bright field photos in Figure 2 and Figure 5? It would be informative to specify the regions.
3. Authors may describe the potential translation and possible limitations of the platform in the discussion section.

Reviewer #2 (Remarks to the Author):

In this MS the authors use lipid nanoparticles containing modified PEG to assess the delivery of RNA molecules to the photoreceptor cells in the mouse retina. They show that carboxy modifications to the LNP lead to a greater RNA delivery to the photoreceptors than unmodified or amino-modified LNPs. While the extent of the delivery is still limited, these results are of interest as a potential method for the safe delivery of gene editing tools. There are, however, aspects to the manuscript that need improvement.

1. The focus of the work is to use this system for the delivery of gene editing tools to the photoreceptor cells for therapeutic use. The authors acknowledge that the editing in photoreceptor cells was 'low level', but the levels that were achieved are not actually reported anywhere. Fig 5H shows that the authors have managed to edit at least 2 photoreceptors after dosing between 6 and 9 eyes. Were these the only edited cells they found? If not, what was the average number of red photoreceptor cells per eye?

2. The authors express almost all quantification as fold-change over controls. While that can be reasonable when the raw values do not mean anything, it does remove potentially valuable data when the raw values carry interpretable information. The most jarring occasion is Figure 3C, where the authors show that the number of photoreceptors co-expressing tdTomato and recoverin has doubled. That relative change is unfortunately not very informative. What the readers would like to know is what proportion of the photoreceptors they managed to target – did the co-localisation increase from 1% to 2% of cells, or from 40% to 80%? The former would leave a lot of work to be done, the latter would be an amazing success.

3. The methodology for calculating the fold-change appears to be erroneous, as all the control values are presented as exactly 1 without any variation. The authors should calculate the mean of the controls and then normalise the individual control values against that mean, thus preserving the variability that will presumably have been present in the control population.

4. The quality of the sections is not great, especially in Figure 2. What happened to the nuclear counterstain in 2E and F? It seems to have turned into a blue rinse, which is presumably not due to the LNPs because later sections with those two LNPs have normal nuclei. The red channel in Fig 5F appears to be overexposed making the interpretation of the image difficult. Fig 2K is described as showing “thinning of the ONL”, which is quite an understatement, as there are virtually no photoreceptors left, yet the corresponding image 2D does have a reasonably normal ONL. How do the authors explain that difference? What are the big red blobs in Fig 2D?

5. The value of the Ai9 line is that it will give robust expression in all cells where recombination has occurred, but that mouse line is known to have some readthrough of tdTomato in unrecombined cells. For that reason, it is not clear whether the faint signal in some cells (e.g. the green arrow in Fig 2E) is really a recombination event, when comparing that to the brightly red cone photoreceptors (yellow arrow).

6. It is curious that in Fig 2E almost all cones (and very few rods) appear to be transfected by LNPx, but in Fig 3 that preference for cones appears to be absent. It is even more curious that in Figure S5, it is LNPz which appears to transfect cones preferentially, but not in Fig 2F. How do the authors explain the differences in photoreceptor transfection pattern? The authors should perform co-staining with specific photoreceptor markers to back up the transfection specificity pattern that is suggested by the localisation of the signal.

7. The widespread use of the word ‘expression’ is inaccurate. Expression should refer to the process of transcription of a gene and production of the encoded protein. It is problematic when the word expression is used to refer to a measurement of the protein phenotype. E.g. the authors claim to be measuring ‘tdTomato expression’ in the fundus or on sections (Fig 2 and others). What they are

measuring is fluorescence of course, not expression. How much of that fluorescence is derived from tdTomato and how much from autofluorescence in the retina has not been determined, and to what extent the expression level of the tdTomato protein has changed in the retina is thus unclear. In figure 5I, the image clearly shows that all the tdTomato signal in the ONL is derived from the Müller cell processes traversing that layer. Presenting this as a measurement of “tdTomato expression in the ONL” is misleading, as it will lead readers to equate it to tdTomato being expressed in the photoreceptors and thus photoreceptors having been edited.

8. Page 9-10 “the LNPx variant containing the DLin-MC3-DMA ionizable lipid with unmodified guides led to retinal toxicity in these studies (data not shown)”

Apart from the question whether not showing data is acceptable at a time when there is unlimited supplementary data space, how do the authors explain the sudden presence of toxicity when that same lipid composition was deemed safe in the previous Figures? Is such an unpredictability of toxicity a concern for future use, especially in the clinic?

9. Why did the authors separately measure fluorescence in the outer segments and in the ONL after Cre delivery (Fig 2G and 2H). The tdTomato that is present in both locations is obviously derived from the same source, before spreading to various parts of the cell. A measurement at a single location is as meaningful as at two or three. The ONL is probably more reliable in that respect, as it is more easily demarcated, and less likely to be affected autofluorescence and will not be affected by tdTomato in RPE microvilli that surround tips of the outer segments. From the image in Fig 2G it appears that the measurement area includes both the photoreceptor inner segments and the outer segments. Therefore, if the authors wish to maintain that graph, they should either amend the description, or explain how they managed to determine the border between inner- and outer segments to measure fluorescence from outer segments only and exclude RPE microvilli.

10. Was administration of the various formulations randomised between animals and between eyes? Were the researchers masked to the formulation each individual eye received? Please describe the randomisation/masking measures (or their absence) in the Methods section.

We appreciate the opportunity to revise our manuscript. Addressing the feedback has greatly improved the quality of the manuscript. We'd like to highlight two main areas of improvement. 1) We repeated the Cre mRNA injections in the Ai9 mice with LNP variants. At 7-day post-injection, we have co-stained the retinal sections with visual arrestin, which labels the cytoplasm of rod and cone cell bodies as well as the outer segments (Figure 3) and find very strong localization. We counted the number of tdTomato+ nuclei in the outer nuclear layer, providing a % photoreceptor transfection efficiency for all LNPs evaluated (Figure 2H). This method of quantification has improved the results of this manuscript. 2) We repeated the gene editing injections comparing Cas9-sgAi9-LNPx delivery to a non-targeted Cas9-sgGFP-LNPx control. We generated RPE flatmounts and quantified the area of tdTomato+ RPE in the treated retina (Figure 5F&G). Additional RPE samples went for Next Generation Sequencing (NGS) to check editing percentage in RPE after (Figure 5H). With these two methods, we are able to present a quantification analysis of gene editing events in the retina. Additional experiments were also performed to address other issues raised by the reviewers. We hope these new results and figures address issues raised by the reviewers.

Here is a point-by-point response to each reviewer comments.

Reviewer 1

In this manuscript, Sahay and co-authors conducted an exciting and comprehensive study using lipid nanoparticles (LNPs) for mRNA delivery and gene editing in the eye. The results from two different animal models showed that surface-modified LNPs, specifically those with carboxylation LNP-mRNA formulations exhibited significantly higher expression in the retina and can be applied to target multiple types of retinal cells. Additionally, in vivo results displayed that the modified LNPs encapsulating Cas9 mRNA and sgRNA can effectively induce retinal gene editing. Overall, this is a very meaningful work and an important report on modified LNPs for retinal gene editing. Below are minor comments.

We thank you for your time, expertise, and commitment to improving our research. We have carefully addressed all your concerns and made the necessary changes.

Reviewer 1-1

What is the rationale to choose 0.3% of functionalized PEG as the molar ratio?

In our previous publication, we tested wide range of functional PEG substitution from 0.15% to 1.2% of 1.5% total PEG-lipid content. Optimal transfection resulted with the use of 0.3% functional PEG substitution. Over 0.3% showed decreased levels of transfection. This optimized ratio can be used for different conjugation techniques. Overall, our previously published work supports this ratio of functionalized PEG substitution.

Herrera-Barrera, M. et al. Peptide-guided lipid nanoparticles deliver mRNA to the neural retina of rodents and nonhuman primates. Sci. Adv. 9, eadd4623 (2023).

Reviewer 1-2

What is the location of the bright spot in the bright field photos in Figure 2 and Figure 5? It would be informative to specify the regions.

We have outlined the treated area in the bright field fundus images in Figure 2 and Figure 5. We have also added an arrowhead pointing out the optic nerve head.

Reviewer 1-3

Authors may describe the potential translation and possible limitations of the platform in the discussion section.

We thank the reviewer for these valuable suggestions. We hope to translate this platform into a gene editing therapy for IRD. Currently, EDIT-101 is the most advanced gene editing platform for retinal degeneration. It uses an AAV to deliver the CRISPR/Cas9, our data suggests that we will be able to replace the use of AAVs with LNPs. Limitations of our platform could be transfection efficiency, however, LNPs are able to be re-dosed. Future studies will aim to work out dosing and delivery for optimal editing events. Another limitation is that the LNP we characterized in this study has yet to deliver gene editors to the photoreceptors. Future studies will continue to optimize our LNP platform for photoreceptor delivery of gene editing machinery. We have added following paragraphs in the result sections:

Line 309-311, Page 13: *One main limitation of our carboxylated LNPs was that they were not able to transfect the photoreceptors with gene editing cargo. However, we now have two studies showing that surface modifications are key to altering cellular tropism in the retina.*

Line 331-332, Page 13: *Our goal is to develop translatable LNP-mRNA gene editing systems for IRDs, such as LCA10, RPE specific mutations, autosomal dominant RP and Ushers Syndrome.*

Reviewer 2

In this MS the authors use lipid nanoparticles containing modified PEG to assess the delivery of RNA molecules to the photoreceptor cells in the mouse retina. They show that carboxy modifications to the LNP lead to a greater RNA delivery to the photoreceptors than unmodified or amino-modified LNPs. While the extent of the delivery is still limited, these results are of interest as a potential method for the safe delivery of gene editing tools. There are, however, aspects to the manuscript that need improvement.

Thank you for taking the time to review our manuscript and for providing valuable feedback on our work. We sincerely appreciate your effort in assessing our submission and your insightful comments. Your suggestions and comments have been instrumental in improving the quality of our research. We have carefully considered each of your points and have made the necessary revisions accordingly.

Reviewer 2-1

The focus of the work is to use this system for the delivery of gene editing tools to the photoreceptor cells for therapeutic use. The authors acknowledge that the editing in photoreceptor cells was 'low level', but the levels that were achieved are not actually reported anywhere. Fig 5H shows that the authors have managed to edit at least 2 photoreceptors after dosing between 6 and 9 eyes. Were these the only edited cells they found? If not, what was the average number of red photoreceptor cells per eye?

We appreciate the reviewer's feedback on our study regarding the low number of edited photoreceptor cells. It is true that we observed only a small number of photoreceptor cells being edited, typically ranging from 2 to 3 cells per eye (Now Figure S6F-G). However, we observed editing of both retinal pigment epithelium (RPE) cells and Müller glia cells. To address reviewer's concern, we conducted additional experiments using Ai9 mice and performed flat-mount imaging of the RPE (Figure 5F) and neural retina (Figure S6D). The expression of edited RPE cells is shown in Figure 5F, while the editing of Müller glia cells is highlighted in Figure S6E. As the CRISPR-Cas9 editing was mostly seen in RPE, we have quantified the tdTomato positive RPE cells and presented them in Figure 5G.

To further substantiate our findings, we went beyond visual confirmation and employed next-generation sequencing (NGS) techniques. This allowed us to quantitatively determine the total editing percentage in the RPE cells from entire RPE tissue, which was exhibited 0.51% of reads containing indels, confirming a 5.94-fold higher RPE editing compared to the control group (non-targeted Cas9-sgGFP encapsulated LNPx) (Figure 5H). This objective measurement reinforces the efficacy of our editing approach and adds robustness to our results. While we acknowledge the need for improvements in editing photoreceptor cells, we are already actively working on developing a new formulation that will enhance the efficiency of this process. Our goal is to overcome the current challenges and expand our editing capabilities to encompass a larger number of photoreceptor cells. However, this study shows for the first time LNPs intact, can be used for gene editing albeit in RPE. We observe that inclusion of two nucleic acids Cas9 and a sgRNA modified surface properties and reduced PR based editing. On the other hand, LNPx with Cre-mRNA with anionic charge resulted in substantial fluorescence from the photoreceptors the Ai9 model, thus suggesting that surface properties are essential for cell selective delivery.

We have added following paragraphs in the result sections:

Page 10 line 249-255: *At 7 days post-injection, the Cas9-sgAi9 fundus imaging revealed a significantly higher gross tdTomato fluorescence signal compared to the Cas9-sgGFP group,*

with a fold change of 1.97 (** $p < 0.01$) (Figures 5D-E). Upon necropsy, RPE and neural retina flat mounts were generated. Imaging and quantification of editing events in RPE retinal flatmounts resulted 16.4% tdTomato positive RPE cells within the treated area compared to the control group (**** $p < 0.0001$) (Figures 5F-G, & S6C). The examination neural retina flatmounts and retinal cross-sections confirmed the editing of Müller glia cells (Figures S6D-E).

Page 11 line 257-263: To strengthen the findings, NGS techniques were employed by extracting genomic DNA from RPE cells. This allowed for the quantitative determination of the total editing percentage in the RPE from high-quality NGS data (Figure S7). The Cas9-sgAi9 group exhibited 0.51% of reads containing indels, confirming a 5.94-fold higher RPE editing compared to the control group (Figure 5H). Overall, the results indicated that Cas9-sgAi9 delivered by LNPx mediated gene editing in the retina, with the most robust editing observed in the RPE.

Details of the NGS sequencing can be accessed through BioProject: PRJNA991562.

Link:

<https://dataview.ncbi.nlm.nih.gov/object/PRJNA991562?reviewer=nvkmd77aja8jvqogd2q0rf6548&archive=sra>

Reviewer 2-2

The authors express almost all quantification as fold-change over controls. While that can be reasonable when the raw values do not mean anything, it does remove potentially valuable data when the raw values carry interpretable information. The most jarring occasion is Figure 3C, where the authors show that the number of photoreceptors co-expressing tdTomato and recoverin has doubled. That relative change is unfortunately not very informative. What the readers would like to know is what proportion of the photoreceptors they managed to target – did the co-localisation increase from 1% to 2% of cells, or from 40% to 80%? The former would leave a lot of work to be done, the latter would be an amazing success.

We have taken the reviewer's valid feedback into account regarding the limitations of using fold-change over controls for quantification. We observed more consistent and robust results with confocal imaging than flow cytometry experiments (that we since repeated multiple times). This difference can be attributed to the distinct methodologies used for imaging. In confocal microscopy, retinal sections are imaged specifically from the bleb region. However, during flow cytometry, photoreceptors are collected from the entire retina, which results in a dilution of cells from the specific region of interest. We also observed that loss of lot of photoreceptors cells due to multiple staining and washing steps. Due to lack of clarity using the flow cytometry we decided to remove that data and focus on enhancing our robustness through co-staining with visual arrestin, which labels the cytoplasm of rod and cone cell bodies. We also observe a very strong localization with outer segments (Figure 3A-C).

We also focused on improved quantification in the retinal cross sections (Figure 2) and used two new techniques to measure (NGS and flat mount based imaging) for the gene editing experiments in Figure 5.

We have added following paragraphs in the result sections:

Page 8, line 189-192: *Quantification of tdTomato-positive photoreceptors in the ONL revealed that LNPx treatment resulted in 26.9% tdTomato-positive photoreceptors (**** $p < 0.0001$), followed by LNPz with 16.5% (* $p < 0.05$), LNPa with 3.26% and unmodified LNP with 0.06% (Figure 2H).*

Page 8, line 199-203 *To further confirm our results of improved photoreceptor transfection, we performed immunofluorescence (IF) analysis using visual arrestin, which labels the cytoplasm of rod and cone cell bodies as well as the entire outer segment. Staining with visual arrestin (as well as tdTomato images in Figure 2E&F) show that LNPx and LNPz enable transfection of both rod and cones (Figure 3A-C) and a substantial localization with the outer segments.*

Reviewer 2-3

The methodology for calculating the fold-change appears to be erroneous, as all the control values are presented as exactly 1 without any variation. The authors should calculate the mean of the controls and then normalise the individual control values against that mean, thus preserving the variability that will presumably have been present in the control population.

We appreciate your suggestion, and we have implemented the recommended modification in our analysis. As per your advice, we have calculated the average of the control values and normalized each individual value relative to that average. This normalization procedure has been applied to necessary figures throughout our study.

Reviewer 2-4

The quality of the sections is not great, especially in Figure 2. What happened to the nuclear counterstain in 2E and F? It seems to have turned into a blue rinse, which is presumably not due to the LNPs because later sections with those two LNPs have normal nuclei. The red channel in Fig 5F appears to be overexposed making the interpretation of the image difficult. Fig 2K is described as showing “thinning of the ONL”, which is quite an understatement, as there are virtually no photoreceptors left, yet the corresponding image 2D does have a reasonably normal ONL. How do the authors explain that difference? What are the big red blobs in Fig 2D?

We appreciate your attention to the matter. Due to focusing issues with the confocal microscope, the DAPI nuclear counterstain (blue) in Figures 2E and F was not clear. However, we have now re-injected and imaged the retinal sections, and the nuclear counterstain is clearly visible. Additionally, we have addressed the problem with the overexposed image (Figure 5F, previous figure) by capturing new flat mount images after conducting a new injection in the Ai9 mice (Now Figure 5F and S6D).

In our observations of the LNPa injected group, we noticed thinning of the outer nuclear layer (ONL) in certain eyes and this toxicity could be associated with the

positive surface potential of the particles. Moreover, it is important to note that Figures 2D and 2K (previous figures) may have been captured from different regions of the retina, resulting in distinct morphologies. To address this, we have re-imaged all the retinal sections, ensuring they are properly focused and facilitating easier comparisons (Now Figure 2, panel D). The presence of large blobs in Figure 2D (previous figure) may have been caused by broken tissue overlap during the cryosectioning process. We acknowledge this issue, and we have taken steps to rectify it in the newly presented figure (Figure 2D, current figure).

Reviewer 2-5

The value of the Ai9 line is that it will give robust expression in all cells where recombination has occurred, but that mouse line is known to have some readthrough of tdTomato in unrecombined cells. For that reason, it is not clear whether the faint signal in some cells (e.g. the green arrow in Fig 2E) is really a recombination event, when comparing that to the brightly red cone photoreceptors (yellow arrow).

We acknowledge the reviewer's concern about possible tdTomato autofluorescence originating from unrecombined cells. In order to reduce this background signal, we adopted a strategy where we first established the optimal imaging parameters using the PBS-injected Ai9 eye samples. Subsequently, we exclusively imaged the LNP-injected groups. We believe that this background correction method, utilizing PBS-injected Ai9 mouse eyes, effectively minimizes the potential interference from tdTomato signal originating from unrecombined cells.

Additionally, for all experiments in the Ai9 mouse, we added controls including: PBS, unmodified LNP and nontargeted LNP, to compare our experimental groups with. Thus, if there was background tdtomato it would be captured in these groups and accounted for when making comparisons.

Reviewer 2-6

It is curious that in Fig 2E almost all cones (and very few rods) appear to be transfected by LNPx, but in Fig 3 that preference for cones appears to be absent. It is even more curious that in Figure S5, it is LNPz which appears to transfect cones preferentially, but not in Fig 2F. How do the authors explain the differences in photoreceptor transfection pattern? The authors should perform co-staining with specific photoreceptor markers to back up the transfection specificity pattern that is suggested by the localisation of the signal.

We agree with the reviewer that there is some eye-to-eye variability in transfection of rods and cones cells both in LNPx and LNPz variants. Overall, our findings indicate that both LNPx and LNPz are able to transfect both rods and cones. Our updated figures (Figure 2E & 2F) provide a clearer representation between the treated groups. We have performed the co-staining with the visual arrestin and recoverin which further confirm the tdTomato signal localization (Figure. 3A-C & 3E).

We have added following statement in the result sections:

Page 8, line 186-189: *LNPx and LNPz injections resulted in robust tdTomato fluorescence signal in both photoreceptors and the RPE. With LNPx and LNPz treatment, strong tdTomato fluorescence was observed in the outer segments, inner segments, outer nuclear layer, and the synaptic region of photoreceptors (Figure 2E-F).*

Page 8, line 201-203: *Staining with visual arrestin (as well as tdTomato images in Figure 2E&F) show that LNPx and LNPz enable transfection of both rod and cones (Figure 3A-C) and very strong localization with the outer segment.*

Reviewer 2-7

The widespread use of the word 'expression' is inaccurate. Expression should refer to the process of transcription of a gene and production of the encoded protein. It is problematic when the word expression is used to refer to a measurement of the protein phenotype. E.g., the authors claim to be measuring 'tdTomato expression' in the fundus or on sections (Fig 2 and others). What they are measuring is fluorescence of course, not expression. How much of that fluorescence is derived from tdTomato and how much from autofluorescence in the retina has not been determined, and to what extent the expression level of the tdTomato protein has changed in the retina is thus unclear. In figure 5I, the image clearly shows that all the tdTomato signal in the ONL is derived from the Müller cell processes traversing that layer. Presenting this as a measurement of "tdTomato expression in the ONL" is misleading, as it will lead readers to equate it to tdTomato being expressed in the photoreceptors and thus photoreceptors having been edited.

Thank you for your suggestions. In order to provide greater clarity, we have replaced the term "expression" with either "fluorescence signal" or "intensity" in the relevant sections. Furthermore, we have made changes to Figure 2 and other related figures by replacing the tdTomato expression legend on the Y-axis with the mean fluorescence intensity. In addition, we have performed measurements of the total tdTomato autofluorescence in the PBS-treated group and subtracted this value from the corresponding LNP variant-treated groups. This subtraction enables a more accurate assessment of the net tdTomato fluorescence intensity, thereby facilitating a comprehensive determination of the total tdTomato protein expression in the retina. We hope these adjustments contribute significantly to enhance the understanding of our study results. While we have modified expression to fluorescence there are a few places where we have kept expression within the legends since successful delivery of Cre mRNA does lead the genetic recombination and expression of tdTomato that in turn produces a protein. For ease for our readers, we kept in some expression in certain places but by an large have modified based on the input of the reviewer.

In response to the reviewer's feedback, we have taken the decision to exclude Figure 5I (previous ONL quantification graph) from our presentation. This figure previously depicted the measurement of tdTomato fluorescence signals specifically from

the outer nuclear layer (ONL). However, we agree with the reviewer that its inclusion could potentially lead readers to believe that photoreceptors express tdTomato.

Instead, we have changed our focus to the RPE. We generated RPE flatmounts where we quantified the area of tdTomato+ RPE cells in the treated retina (Figure 5F&G). We also introduced Next Generation Sequencing (NGS) data in Figure 5H to provide a more comprehensive understanding of our findings. This NGS data reveals the total editing percentage occurring in the isolated RPE tissue. Our analysis shows a total editing percentage of 0.51%, representing the proportion of reads containing indels and substitutions at the cut site. This NGS data offers valuable insights into the overall effectiveness of the editing process. Details of NGS method has been included in the method section.

We have added following statement in the result sections:

Page 11 line 257-263: *To strengthen the findings, next generation sequencing (NGS) techniques were employed by extracting genomic DNA from RPE cells. This allowed for the quantitative determination of the total editing percentage in the RPE from high-quality NGS data (Figure S7). The Cas9-sgAi9 group exhibited 0.51% of reads containing indels, confirming a 5.94-fold higher RPE editing compared to the control group (Figure 5H). Overall, the results indicated that Cas9-sgAi9 delivered by LNPx mediated gene editing in the retina, with the most robust editing observed in the RPE.*

Reviewer 2-8

Page 9-10 “the LNPx variant containing the DLin-MC3-DMA ionizable lipid with unmodified guides led to retinal toxicity in these studies (data not shown)”

Apart from the question whether not showing data is acceptable at a time when there is unlimited supplementary data space, how do the authors explain the sudden presence of toxicity when that same lipid composition was deemed safe in the previous Figures? Is such an unpredictability of toxicity a concern for future use, especially in the clinic?

As per reviewer concern, we have presented the Cas9-sgRNA encapsulated DLin-MC3-DMA-LNPx toxicity figure in supplementary section (Figure S6B). The sudden presence of toxicity could be payload-specific effect. The Cas9-sgRNA payload itself can impact cellular processes and trigger immune responses. The combination of MC3 lipid with Cas9-sgRNA may result in unexpected interactions, enhancing toxicity which didn't observe in smaller length cargo. Regarding the clinical implications, the unpredictability of toxicity observed with the MC3 lipid could raise concerns for future use in clinical settings especially for larger payload delivery. Use of biodegradable lipids can significantly decrease the toxicity so here we used LP01. However, safety evaluations and toxicity assessments should be thoroughly tested before considering the translation of lipid-based delivery systems into clinical applications.

Reviewer 2-9

Why did the authors separately measure fluorescence in the outer segments and in the ONL after Cre delivery (Fig 2G and 2H). The tdTomato that is present in both locations

is obviously derived from the same source, before spreading to various parts of the cell. A measurement at a single location is as meaningful as at two or three. The ONL is probably more reliable in that respect, as it is more easily demarcated, and less likely to be affected autofluorescence and will not be affected by tdTomato in RPE microvilli that surround tips of the outer segments. From the image in Fig 2G it appears that the measurement area includes both the photoreceptor inner segments and the outer segments. Therefore, if the authors wish to maintain that graph, they should either amend the description, or explain how they managed to determine the border between inner- and outer segments to measure fluorescence from outer segments only and exclude RPE microvilli.

We agree with the reviewer that the tdTomato fluorescence observed in both the outer segment and outer nuclear layer (ONL) after Cre delivery originates from the same source. We have replaced these data with new figures that quantify tdTomato-positive photoreceptor cells specifically in the ONL (now Figure 2H).

10. Was administration of the various formulations randomised between animals and between eyes? Were the researchers masked to the formulation each individual eye received? Please describe the randomisation/masking measures (or their absence) in the Methods section.

There were no randomisation/masking measures used in the study and we have included in the method section.

REVIEWERS' COMMENTS

Reviewer #1 (Remarks to the Author):

In this revised manuscript, the authors provided additional description and information to clarify the experiment procedures and data analysis. Experimental data support the study design and conclusions.

Reviewer #2 (Remarks to the Author):

The reviewers' comments have been addressed well and the manuscript is now much improved. The study will be of wide interest to the field of gene therapy.

We would like to express our sincere gratitude for your positive feedback on our manuscript. We are thrilled to inform you that we have diligently worked on addressing all of the editorial concerns and suggestions provided during the peer review process. Your guidance and the valuable feedback from the reviewers have been instrumental in refining our work. Most of the author checklist concerns are addressed, and we hope that the final version of the manuscript now meets the high standards of excellence upheld by this journal.

Reviewer 1

In this revised manuscript, the authors provided additional description and information to clarify the experiment procedures and data analysis. Experimental data support the study design and conclusions.

We wanted to extend our heartfelt gratitude for your invaluable feedback and guidance throughout the review process.

Reviewer 2

The reviewers' comments have been addressed well and the manuscript is now much improved. The study will be of wide interest to the field of gene therapy.

We thank the reviewer for the positive comment. Your expertise and constructive comments were instrumental in improving the quality of our manuscript. We genuinely appreciate your time and effort in helping us refine our work.